# Common and Differential Dynamics of the Function of Peripheral Blood Mononuclear Cells between Holstein and Jersey Cows in Heat-Stress Environment

**DOI:** 10.3390/ani11010019

**Published:** 2020-12-24

**Authors:** Eun Tae Kim, Sang Seok Joo, Dong Hyeon Kim, Bon-Hee Gu, Da Som Park, Rahman Md Atikur, Jun Kyu Son, Beom Young Park, Sang Bum Kim, Tai-Young Hur, Myunghoo Kim

**Affiliations:** 1Dairy Science Division, National Institute of Animal Science, Rural Development Administration, Cheonan 31000, Korea; etkim77@korea.kr (E.T.K.); kimdh3465@korea.kr (D.H.K.); junkyuson@korea.kr (J.K.S.); tyohur@korea.kr (T.-Y.H.); 2Department of Animal Science, College of Natural Resources & Life Science, Pusan National University, Miryang 50463, Korea; ssjoo7680@gmail.com (S.S.J.); parkdasome@gmail.com (D.S.P.); md.rahman@bau.edu.bd (R.M.A.); 3Life and Industry Convergence Research Institute, Pusan National University, Miryang 50463, Korea; g.bonhee@gmail.com; 4National Institute of Animal Science, Rural Development Administration, Wanju 55365, Korea; byp5252@korea.kr; 5Rural Development Administration, Jeonju 54875, Korea; sangbkim@korea.kr

**Keywords:** immune response, Holstein cow, Jersey cow, transcriptome, THI

## Abstract

**Simple Summary:**

Seasonal change, particularly changing to hot and humid season, has a negative effect on dairy cows in various ways, including productivity, reproduction, metabolism, and immunity. In high-temperature and humid weather, dairy cows are vulnerable to diseases by weakened immune system. However, the cause of this has not been fully described. Therefore, this study aims to understand changes of specific gene expression and immune pathways based on transcriptome analysis from peripheral blood mononuclear cells of Holstein and Jersey dairy cows between normal and heat-stress environmental conditions. We observed that the two breeds of dairy cow have common and different immune shifts according to the changes of temperature and humidity condition. Overall, the findings of this study improve the understanding of the underlying mechanisms by which seasonal changes affect dairy cow immunity.

**Abstract:**

Heat stress has been reported to affect the immunity of dairy cows. However, the mechanisms through which this occurs are not fully understood. Two breeds of dairy cow, Holstein and Jersey, have distinct characteristics, including productivity, heat resistance, and disease in high-temperature environments. The objective of this study is to understand the dynamics of the immune response of two breeds of dairy cow to environmental change. Ribonucleic acid sequencing (RNA-seq) results were analyzed to characterize the gene expression change of peripheral blood mononuclear cells (PBMCs) in Holstein and Jersey cows between moderate temperature-humidity index (THI) and high THI environmental conditions. Many of the differentially expressed genes (DEGs) identified are associated with critical immunological functions, particularly phagocytosis, chemokines, and cytokine response. Among the DEGs, *CXCL3* and *IL1A* were the top down-regulated genes in both breeds of dairy cow, and many DEGs were related to antimicrobial immunity. Functional analysis revealed that cytokine and chemokine response-associated pathways in both Holstein and Jersey PBMCs were the most important pathways affected by the THI environmental condition. However, there were also breed-specific genes and pathways that altered according to THI environmental condition. Collectively, there were both common and breed-specific altered genes and pathways in Holstein and Jersey cows. The findings of this study expand our understanding of the dynamics of immunity in different breeds of dairy cow between moderate THI and high THI environmental conditions.

## 1. Introduction

As a consequence of global climate change, rapid environmental alterations have exerted significant influences on the fields of agriculture, including livestock. Among domestic animals, high-productivity dairy cows are most sensitive to heat stress [1]. At high ambient temperatures, dairy cows show a reduction in milk yield and quality, as well as in reproductive performance, resulting in significant economic loss to the dairy industry [2]. Thus, heat stress is a major concern that should be addressed to maintain the high productivity of dairy cows [3].

It has been reported that dairy cows in high-temperature and humidity environments are usually unhealthy, suffering from various infectious and metabolic diseases [4]. Correlation studies have revealed that heat-stressed dairy cows have a higher incidence of diseases, such as bovine respiratory disease and mastitis [5,6]. Poor health status due to heat stress can be primarily explained by the immune system imbalance that it induces. Many studies have suggested that heat stress increases disease susceptibility of dairy cows through weakened immunity [2,7]. For example, heat-stressed dairy cows show abnormal innate and adaptive immune responses, resulting in an attenuated vaccine response [3,8]. Although the exact mechanism is unclear, it is commonly hypothesized that heat-stress-induced glucocorticoids negatively impact immune homeostasis through altered cytokine production [3,9]. 

Holstein and Jersey cows are two major breeds of dairy cow [10]; they have distinct characteristics in terms of productivity and heat resistance [11,12]. Holstein cows are known to produce more milk than Jersey cows; however, Jersey milk has a higher content of milk solids (milk fat and protein) than Holstein milk [13]. In terms of heat resistance, it was reported that respiration and pulse rates were similar in Holstein and Jersey cows in a normal environment, but Jersey cows had lower body temperatures compared with Holstein cows under heat-stress conditions [14]. In addition, the reduction in milk yield in Jersey cows was less than the response in Holstein milk yield during high ambient temperatures [15]. Thus, it is generally accepted that Jersey cows have a greater capacity to adapt to higher ambient temperatures. 

There are a few studies that investigate immune changes of Holstein and Jersey cows. In a grain challenge study, systemic and molecular biomarkers for inflammatory, stress, and metabolic responses were compared between Holstein and Jersey cows [16]. In brief, Holstein cows had relatively more altered biomarkers of inflammation and stress such as cytokines, heat shock proteins, and oxidative stress genes after the grain challenge. Another study showed transcriptome changes in whole blood in Holstein and Jersey calves induced by weaning treatment [17]. Through the analysis of blood samples pre- and post-weaning, Johnston et al. demonstrated differential immune responses and transcriptional activity of genes in whole blood in Holstein and Jersey calves induced by weaning stress. Although several studies have shown differential productivity and physiological changes due to heat stress in Holstein and Jersey cows, we do not understand how these two breeds of dairy cow immunologically respond to heat stress. 

To understand the changes in the immunophysiological response of the two breeds of dairy cow induced by THI changes, we compared the peripheral blood mononuclear cell (PBMC) transcript profiles of Holstein and Jersey cows under moderate and high temperature-humidity environmental conditions. We attempted to identify common and breed-specific heat-stress-sensitive genes and pathways in the immune cells. We identified cytokine and chemokine responses as the most important immune pathways sensitive to hot and humid weather in both Holstein and Jersey dairy cows. Furthermore, we also identified breed-specific genes and pathways from this study.

## 2. Materials and Methods

### 2.1. Experimental Animals 

All procedures and experimental protocols for the animal experiment were conducted as approved by the Institutional Animal Care and Use Committee of the National Institute of Animal Science (NIAS), Rural Development Administration, Republic of Korea (Approval Number: NIAS-2019107). The principles of laboratory animal care were met, and blood collection procedures were performed in accordance with the guidelines. The study was conducted from May to August 2019 at the NIAS on cows in their herd.

Five Holstein cows (56 ± 1.5 months old; body weight 665 ± 31.5 kg; parity numbers 2.4 ± 0.5) and five Jersey cows (61 ± 3.5 months old; body weight 478 ± 20.9 kg; parity numbers 2.4 ± 0.8) were used for the experiments. The average milk yields of Holstein cows and Jersey cows were 38.5 ± 4.5 kg/day and 30.4 ± 2.6 kg/day in May, respectively. In August, the average milk yields of Holstein cows and Jersey cows were 30.8 ± 5.3 kg/day and 24.2 ± 3.2 kg/day, respectively. Holstein and Jersey dairy cows used in this study had similar days in milk. Holstein cows averaged days in milk (DIM) was 139 ± 8.0 and Jersey cows averaged DIM was 128 ± 15.0 in moderate THI (MT) environment, respectively. All Holstein cows had been part of dairy herd in the NIAS, whereas all Jersey cows were born by embryo transfer to Holstein surrogate mothers and had been bred up for over 40 months under the same management procedures in the NIAS. Vaccination for dairy cows was performed with bovine ephemeral fever virus, akabane virus, infectious bovine rhinotracheitis, bovine virus diarrhea, parainfl uenza-3, bovine respiratory syncytial virus, Anthrax–Blackleg combined vaccine according to the vaccination protocol of the farm. All vaccinations were completed one month before the start of the experiment. All experimental dairy cows were housed in individual stalls and fed total mixed ration (TMR) ad libitum twice daily. A total of 20 kg of TMR was offered at 09:00 h and 16:00 h to each cow, and there was no residual TMR in any of the dairy cows. TMR was formulated to meet or exceed the energy requirements, and to preclude the selection of dietary components. The barn was equipped with a fan ventilation system. The working time of this system was from at 09:00 h to at 18:00 h. Appendix A Presents the ingredients and nutrients of the experimental diet.

### 2.2. Measurement of THI, Respiration, and Rectal Temperature

To determine the effects of THI condition on dairy cows, two distinct annual periods based on temperature-humidity index (THI), moderate THI (MT) and High THI (HT) season, were used for the comparative experiment. Temperature and humidity were recorded every day at 14:00 h for a week using a Testo 174H Mini temperature and humidity data logger (Testo Korea Ltd., Seoul, Republic of Korea). We used the THI equation devised by the National Research Council [18]: THI = (1.8 × ambient temperature + 32) − [(0.55 − 0.0055 × relative humidity) × (1.8 × ambient temperature − 26)]. The average THI for the MT condition in the first week of May was 67.8 ± 5.0 [(average 21.7 °C and 61.2% relative humidity (RH))], and for the HT condition in the second week of August, it was 87.3 ± 1.3 (average 32.8 °C and 79.4% RH). For the MT condition, we collected samples on May 3rd, THI = 69.6 (24.4 °C and 36.3% RH), and for the HT condition on August 9th, THI = 87.5 (35.5 °C and 59.6% RH). All blood samples were collected after monitoring THI.

Respiration rates were measured by counting flank shifts during a 15-s interval and multiplied by 4 to obtain breaths per minute. Rectal temperatures were measured using a standard digital thermometer (KD-133, Polygreen CO., Ltd., Stahndorf, Germany). To reduce possible variations, each measurement was taken twice, and the average value was used. The respiration rate and rectal temperature of dairy cows were recorded before blood collections.

### 2.3. Blood Collection and PBMC Isolation

Blood samples were collected from the jugular vein of the dairy cows into 10 mL K2-EDTA vacutainer tubes (BD Vacutainer, Becton Dickinson Co., Franklin Lakes, NJ, USA). Two tubes were collected from each animal to obtain 15 to 20 mL of blood, and the samples were immediately placed on ice and transported to the laboratory for the isolation of PBMC within 30 min of the time of sampling.

The PBMCs were isolated using density gradient centrifugation. Briefly, whole blood samples were diluted with phosphate-buffered saline (PBS) in a 1:1 ratio in 15 mL conical tubes. Then, 4 mL of Lymphoprep (STEMCELL Technologies lnc., Vancouver, BC, Canada) was added to the new tubes, and 8 mL of the diluted blood samples were overlaid on the Lymphoprep. After centrifugation for 20 min at 800× *g* and at 22 °C, the layer of cells above the Lymphoprep was collected and washed twice with PBS to obtain purified PBMCs that were then suspended in 1 mL of TRIzol reagent (Invitrogen, Carlsbad, CA, USA) and transferred to a 1.5 mL tube. The PBMCs were immediately stored at −80 °C until the RNA isolation process.

### 2.4. Total RNA Isolation and Sequencing

RNA was isolated from the PBMC samples in TRIzol them, concentration of the total RNA was measured by Quant-IT RiboGreen (Invitrogen, Carlsbad, CA, USA). The integrity of the total RNA was determined by TapeStation RNA screentape (Agilent Technologies, Santa Clara, CA, USA). The RNA with an RNA integrity number value greater than 7.0 were used for the RNA library construction. 1 μg of total RNA was used to generate the RNA library with an Illumina TruSeq Stranded mRNA Sample Prep Kit (Illumina, Inc., San Diego, CA, USA) [19]. Briefly, rRNA was removed from total RNA using the Ribo-Zero rRNA Removal Kit (Human/Mouse/Rat) (Illumina, Inc., San Diego, CA, USA). The remaining mRNA was fragmented into small pieces using divalent cations at an elevated temperature. The cleaved mRNA fragments were copied into first strand cDNA using SuperScript II reverse transcriptase (Invitrogen, Carlsbad, CA, USA) with random primers. Next, second-strand cDNA synthesis using DNA Polymerase I, RNase H, and dUTP. The cDNA fragments were subjected to an end repair process, the addition of a single ‘A’ base, and ligation of the adapters. The final products were then purified and enriched with PCR to generate the final cDNA libraries. The quantification of the libraries was performed by using the KAPA Library Quantification kit for Illumina Sequencing platforms according to the qPCR Quantification Protocol Guide (KAPA Biosystems Inc., Wilmington, MA, USA). The quality was assessed by using the TapeStation D1000 ScreenTape (Agilent Technologies, Santa Clara, CA, USA). All reagents and kits were used according to the manufacturer’s guidelines. Indexed libraries were then sequenced on the Illumina NovaSeq (Illumina, Inc., San Diego, CA, USA), and the paired-end (2 × 100 bp) sequencing was performed by the Macrogen Inc. (Seoul, Republic of Korea).

### 2.5. Data Handling Procedures

RNA-seq analysis of samples included quality control, transcript assembly, abundance estimation, and differential expression analysis. Before analysis, FastQC (version v0.11.7) was used to check the quality of the raw sequence data from high throughput sequencing pipelines, and Trimmomatic (v0.38) was used to trim the data [20]. One PBMC sample from Jersey cows in May was not passed for quality control test, so 5 (MT) and 5 (HT) samples of Holstein PBMC and 4 (MT) and 5 (HT) of Jersey PBMC were used for further analysis. HISTA2 (v2.1.0) and Bowtie2 (v2.3.4.1) were used to align the RNA sequences and to map the sequence. Aligned RNA-seq reads were assembled using the reference gene model as previously described [21]. StringTie (v 1.3.4d) was used to estimate the relative abundance of transcripts with read count values. From the total genes, those with a count of 0 in at least one sample were excluded. The data was corrected to relative log expression for normalization using the DESeq2 R library [22]. To confirm sample-to-sample grouping, principal coordinates analysis (PCoA) was conducted using normalized values with Past 4.03 software.

DEG data were statistically analyzed for functional annotation and pathway analysis. For Gene Ontology (GO) analysis, the g: Profiler in R package was used for enrichment analysis based on the data source [23]. The GO terms analysis associated with three main GO categories, biological process, cellular component, and molecular function, were conducted. The significances of GO terms were calculated by hypergeometric and multiple testing correction to adjusted *p*-value (*p* < 0.05). The DEGs also underwent a pathway analysis using the Kyoto Encyclopedia of Genes and Genomes (KEGG) to analyze the biological pathways most associated with DEGs. The significance of the pathways was determined by the modified fisher’s exact test (*p* < 0.05). Then, the *p*-value was corrected by the false discovery rate (FDR) method to generate the cut-off KEGG pathways (FDR < 0.01). Gene network analysis was performed on the unique DEGs by using STRING software (string-db.org/). The STRING was used to perform analysis to visualize—through gene networks, the protein-protein interactions in a list of unique genes. Prism software 7 (GraphPad, La Jolla, CA, USA) was used for the heatmap visualization of DEGs. A chord diagram in NetworkAnalyst (v3.0) was used for the summarization of Holstein and Jersey cow genes. 

### 2.6. Statistical Analysis

The statistical analyses of respiration rate and rectal temperature were conducted using Prism 7 software. A repeated measures two-way analysis of variance (two-way ANOVA) with Tukey’s multiple comparison test was used. The effect of two breeds (Holstein or Jersey cow) and two THI conditions (MT or HT) were used for statistical analysis. All data are presented as mean ± SD and *p* < 0.05 were considered statistically significant.

## 3. Results

### 3.1. Changes in Respiration Rate and Rectal Temperature of Two Breeds of Dairy Cow by THI Condition

To determine whether Holstein and Jersey cows show physical responses in ambient temperature condition based on THI, we monitored respiration rates and rectal temperatures of dairy cows in MT (THI = 69.6) and HT condition (THI = 87.5). In the HT condition, the respiration rates increased significantly in both Holstein and Jersey cows (48 ± 5.37 breath/min to 89.2 ± 14.12 breath/min, Holstein cows; 58.4 ± 30.21 breath/min to 108 ± 13.15 breath/min, Jersey cows), but there was no significant difference in respiration rates between two breeds of dairy cow in this condition (Figure 1A). However, rectal temperatures increased significantly in Holstein cows in the HT condition but not in Jersey cows (37.98 ± 0.35 °C to 38.82 ± 0.43 °C, Holstein cows; 37.90 ± 0.35 °C to 38.48 ± 0.38 °C, Jersey cows) (Figure 1B). 

### 3.2. Preliminary Analysis and Summary Statistics for RNA Sequencing Data 

For RNA sequencing (RNA-seq) analysis, Holstein PBMC samples [*n* = 5 (MT), *n* = 5 (HT)] and Jersey PBMC samples [*n* = 4 (MT), *n* = 5 (HT)] were used. Before the analysis, raw data were trimmed. Each of the 19 trimmed samples generated an average of 59 million reads. The RNA-seq analysis revealed that an average of 95.57% of the reads were mapped to the annotated bovine reference genome (GCF_002263795.1_ARS-UCD1.2) (Appendix A). After the assembly process, the abundance of each transcript was estimated to read count values. Only previously known transcript expression value (without considering novel transcripts) was used to extract transcript expression values of the samples.

### 3.3. Analysis of Differentially Expressed Genes of Two Breeds of Dairy Cow by THI Condition

After transcript assembly, differentially expressed genes (DEGs) screening process was conducted based on read count values. Also, the samples were submitted to data quality check and normalization process for reliability and sample-to-sample similarity. DEG analysis was performed to compare the differential gene expressions of Holstein and Jersey cows between HT and MT condition. In this study, we considered that genes with greater than 2-fold up- or down-regulated and *p* < 0.05 had significantly different DEGs. With quality check, there were total of 14,405 DEGs in the dairy cow PBMC samples. Normalized values of samples of Holstein and Jersey cows were used for PCoA. The samples were ordered by PCoA according to their global similarity and the PCoA plots showed that the global patterns of the transcriptomes were different between MT and HT conditions in both Holstein and Jersey dairy cows (Figure 2A). In the samples, coordinate 1 and coordinate 2 were shown 26.2% and 17.7% (Holstein cows) and 28.0% and 21.7% (Jersey cows), respectively. Through a hierarchical clustering analysis for a DEG list, the degree of sample similarity by sample was grouped for each gene and visualized by heatmap. Hierarchical clustering of DEGs showed that samples of Holstein and Jersey cows were clustered based on MT and HT conditions (Figure 2B). 

In Holstein and Jersey cows, 163 and 188 significant DEGs (*p* < 0.05, fold-change < −2 or > 2), respectively, were identified between HT and MT condition (Appendix A). A list for up- and down-regulated DEGs for Holstein and Jersey PBMCs reveals that many of the top-ranked DEGs have immune-related functions, including chemokine and cytokine related response. Volcano plots obtained by correcting log value show up- and down-regulated DEGs (Figure 3). The top ten up- and down-regulated DEGs were highlighted. Among down-regulated genes, the expression of genes for *HBA* (−7.01), *HBA1* (−6.83), *KRT17* (−6.12), *CXCL3* (−5.01), *LDLRAD3* (−3.44), *IL1A* (−3.26), *LOC112444653* (−3.17), *LOC112445075* (−2.97), *DAB2* (−2.88)*,* and *LOC112447362* (−2.83) in PBMCs of Holstein cows were down-regulated by the environment change. However, the expression of genes for *GCSAML* (6.70), *LOC112442336* (4.67), *LOC101902491* (4.40), *LOC112449301* (4.11), *LOC112447136* (3.90), *MATN1* (3.78), *CCL8* (3.76), *GPR158* (3.57), *LOC529792* (3.26), and *LOC104975686* (3.19) was significantly increased in heat-stressed Holstein PBMCs.

In Jersey cow PBMCs, we also observed some specific gene expression changes by heat stress. Among down-regulated genes, the genes for *CXCL3* (−17.79), *EDN1* (−10.06), *IL1A* (−9.57), *GPR84* (−9.28), *CXCL2* (−8.98), *GRO1* (−7.80), *CCL4* (−6.68), *LOC107132849* (−5.40), *RN18S1* (−5.10), and *LOC107133284* (−4.69) were significantly decreased in heat-stressed Jersey PBMCs. The genes for *LOC101902555* (6.40), *LOC112442094* (4.57), *LOC112444474* (4.48), *ARFGEF3* (4.34), *LOC104975686* (4.23), *RAMP3* (4.09), *LOC522610* (3.97), *LOC101902491* (3.65), *LOC112448798* (3.43), and *LOC112442541* (3.28) were significantly highly up-regulated in Jersey cow PBMCs by heat stress. Collectively, the *CXCL3* and *IL1A* genes were shown a down-regulated pattern in both Holstein and Jersey cows under HT environment condition, and there were some specific genes that could only be identified in each dairy cow breed.

### 3.4. Functional Annotation of Differentially Expressed Genes of Dairy Cows

The DEG list between HT and MT condition in Holstein and Jersey PBMCs was analyzed for three main GO categories: biological process (BP), cellular component (CC), and molecular function (MF). All GO terms of dairy cows are represented in Appendix A. We identified several overrepresented BP and MF in both dairy cows (*p* < 0.05) (Appendix A). The cellular response to chemokine, chemokine-mediated signaling pathway, and response to chemokine in BP terms were presented in both Holstein and Jersey GO analysis in the heat-stress environment, collectively.

We further investigated the RNA-seq data with significant KEGG to identify the key immunological pathways of PBMCs of dairy cows in response to HT. The 21 pathways and the 68 pathways were significantly different in Holstein and Jersey cows (*p* < 0.05), respectively and they were sorted by FDR. The cut-off KEGG pathways were represented Appendix A (FDR < 0.01). Like GO analysis, Jersey cows represented a more pathway number compared with pathways of Holstein cows, respectively. Among the cut-off KEGG pathways (with the exclusion of the Human Diseases category), the top five significant pathways in Holstein PBMCs were calcium signaling pathway, cytokine–cytokine receptor interaction, viral protein interaction with cytokine and cytokine receptor, cGMP-PKG signaling pathway, and chemokine signaling pathway. Jersey PMBCs included the tumor necrosis factor (TNF) signaling pathway, IL-17 signaling pathway, cytokine–cytokine receptor interaction, viral protein interaction with cytokine and cytokine receptor, and chemokine signaling pathway. It is notable that cytokine–cytokine receptor interaction, viral protein interaction with cytokine and cytokine receptor, and chemokine signaling pathway were commonly presented pathways in Holstein and Jersey cows in the heat-stress environment. The list for all KEGG pathways and the related genes is shown Appendix A.

Among the top-ranked KEGG pathways, the interested pathways are depicted with DEGs in Figure 4. The calcium signaling pathway, the first ranked pathway in heat-stressed Holstein PBMCs, is shown overlaid with gene expression changes in Figure 4A. The expression of several calcium receptor genes, including *CACNA1D* (2.26), *CACNA1E* (2.04), *CXCR4* (2.01), and *P2RX1* (2.13) expression, were elevated in Holstein PBMCs by HT environment. In heat-stressed Jersey PBMCs, the TNF signaling pathway, the first ranked pathway, and the IL-17 signaling pathway, the second ranked pathway, are shown together with gene expression results in Figure 4B. Briefly, down-regulation of genes associated with activator protein 1 (AP-1) and IκBα, resulted in reducing expression of functional molecular genes such as *CXCL2* (−8.98), *CXCL3* (−17.79), and *IL1B* (−2.93). Collectively, GO and KEGG analysis indicated that there were some breed-specific and common pathways such as the cytokine–cytokine receptor interaction and chemokine signaling alterations in Holstein and Jersey cow PBMCs by hot and humid environment.

### 3.5. The Gene Network Analysis of the Unique Differentially Expressed Genes

To determine interactions of DEGs, we performed gene network analysis of DEGs by using STRING (string-db.org/). From the KEGG pathway analysis, we also identified unique pathways, which were significantly changed by hot and humid environment in specific breeds. Figure 5A represents the gene networks of some breed-specific pathways for each of the two types of dairy cows. The calcium signaling pathway and the cGMP-PKG signaling pathway were Holstein-specific pathways. In these pathways, the 5 DEGs, *CACNA1D* (2.26), *CACNA1E* (2.04), *P2RX1* (2.13), *MYLK* (2.15), and *PTAFR* (-2.49) in the calcium signaling pathway, and the 3 DEGs, *CACN1D* (2.26), *MYLK* (2.15), and *RGS2* (2.57) in the cGMP-PKG signaling pathway were the most important interaction partners. Most of the genes associated with these pathways were up-regulated. In Jersey PBMC samples, the 11 DEGs of the TNF signaling pathway including *CXCL2* (−8.98), *CXCL3* (−17.79), *EDN1* (−10.06), Fos proto-oncogene, AP-1 transcription factor subunit (*FOS*) (−2.04), *GRO1* (−7.81), *IL1B* (−2.93), *NFKBIA* (−2.28), *PIK3R3* (−2.34), *ICAM1* (−2.47), *PTGS2* (−2.32), and *TNFAIP3* (−3.90); and the 9 DEGs of the IL-17 signaling pathway, including *CXCL2* (−8.98), *CXCL3* (−17.79), *FOS* (−2.04), *GRO1* (−7.81), *IL1B* (−2.93), *NFKBIA* (−2.28), *PTGS2* (−2.32), *TNFAIP3* (−3.90), and *FOSB* (−2.90), were the most important interaction partners. All genes associated with relevant cytokines pathways decreased by heat stress in Jersey PBMCs.

Gene network analysis was also performed on the cytokine–cytokine receptor interaction and the chemokine signaling pathway, which were common pathways appearing in each of the two breeds of dairy cow. Figure 5B presents the gene interaction for two common pathways with up- or down-regulation of DEGs associated with these pathways. In the Holstein PBMC samples, *CCL8* (3.76), *CXCL2* (−2.70), *CXCL3* (−5.01), *CXCR4* (2.01), *IL1A* (−3.26), *LOC529792* (3.26) (also known as *CRLF2*), and *CCR3* (2.10) were the most important interaction partners. In Jersey PBMC samples, *CCR3* (2.05), *LOC529792* (2.30), *CXCL2* (−8.98), *CXCL3* (−17.79), *GRO1* (−7.81), *IL1B* (−2.93), *CCL3* (−4.40), *CCL4* (−6.68), *IL1A* (−9.57), *IL10* (−3.70), *IL1R2* (−2.41), *NFKBIA* (−2.28), *PIK3R3* (−2.34), and *TNFSF9* (2.63) were the most important interaction partners. Those genes directly or indirectly interacted with several other DEGs. Unlike in Holstein cows, most of the genes in Jersey cows were down-regulated by hyperthermia environment.

### 3.6. Summary of Differentially Expressed Genes and KEGG Pathways in Holstein and Jersey Cows in Response to THI Environmental Condition

Ultimately, we summarized the results of RNA-seq for the comparison of gene expression and identified pathways in PBMCs of the two breeds of dairy cow under HT and MT environment (Figure 6). We found that 163 and 188 genes were significantly altered in Holstein and Jersey cows, respectively, in response to seasonal change. The 41 genes were identified as commonly presented DEGs of PBMCs between each breed of cow analysis (Figure 6A). Among cut-off KEGG pathways of Holstein and Jersey PBMCs, there were common and breed-specific pathways in the top five significant pathways (Figure 6B). Cytokine–cytokine receptor interaction, viral protein interaction with cytokine and cytokine receptor, and chemokine signaling pathway were common KEGG pathways that were significantly altered in both breeds of dairy cow by heat stress. The calcium signaling pathway and the cGMP-PKG signaling pathway were presented in Holstein cows. The TNF signaling pathway and the IL-17 signaling pathway were Jersey-specific pathways in the top five ranked KEGG pathways. There were five genes (CCR3, CXCL2, CXCL3, IL1A, and LOC529792) were identified as common genes and some breed-specific genes associated with top five ranked KEGG pathways in both Holstein and Jersey PBMCs.

## 4. Discussion

In dairy cows, heat stress is known to be directly associated with multiple physiological changes in metabolism, rumen fermentation, growth, lactation, gestation [24], and immune response [25,26]. Heat stress in dairy cows induced by seasonal change usually manifests in an elevated respiration rate, sweating, and an increase in water intake and rectal temperature [27]. Several studies have reported that THI is positively correlated with heat-stress symptoms. For example, in Holstein cows in moderate THI weather, the rectal temperature range was 38.4–38.8 °C in a THI range of 60−72 [28,29]. In contrast, at high THI (72–89), the rectal temperature range was 39.0–39.1 °C [30,31]. However, in hot and humid environments, Jersey cows displayed a milder heat-stress response, with rectal temperatures ranging from 37.2−38.7 °C in a THI range of 70–80 [32].

In our study, the average THI in the first phase of the experiment (moderate THI in the first week of May) was 67.8 ± 2.1 (average 21.7 °C and 61.2% RH), which fits into the “comfort range”. The average THI in the second phase of the experiment (high THI in the second week of August) was 87.3 ± 0.5 (average 32.8 °C and 79.4% RH), which fits into the “moderate heat-stress range”. Both Holstein and Jersey cows in the HT environment displayed symptoms of physiological heat stress, such as increased respiration rate. In this study, Jersey cows showed a milder heat-stress response compared with Holstein cows because rectal temperature increased significantly in Holstein cows but not in Jersey cows; Holstein cows generally had a rectal temperature approximately 0.3 °C higher than Jersey cows in heat-stress conditions. It has been reported that Jersey cows are more resistant to heat stress compared with Holstein cows [33]. Under the hot and humid weather conditions, the decline in milk yield for Holstein cows was more rapid compared with Jersey cows [33,34]. Therefore, our observations concur with previous reports. 

RNA-seq can be a useful tool to produce large amounts of sequence data in non-model organisms such as livestock [35]; for example, it has been used to compare differences between certain breeds of domestic animals, such as dairy cows [17,36]. Several RNA-seq studies have been undertaken to understand the changes in various gene expressions in Holstein dairy cows in a hyperthermic environment. Srikanth et al. reported that genes and signaling pathways for innate immunity were altered in PBMCs in heat-stressed Holstein calves [37]. In addition, Liu et al. conducted transcriptome analysis for whole blood to find potential regulatory genes related to heat tolerance in Holstein cows [38]. They selected heat-tolerant and non-heat-tolerant Holstein cows based on heat-stress scoring parameters such as rectal temperature, respiration rate, and milk yield change, then compared the gene expressions in blood immune cells between heat-tolerant and non-heat-tolerant cows. They identified *OSA*, *MX2*, *IFIT5*, and *TGFB2* genes that are involved in the immune effector process as heat tolerance-related genes. We used PBMCs from two different breeds of dairy cow exposed to THI change to identify common and breed-specific DEGs and pathways. In our study, the gene expression pattern in PBMCs of both Holstein and Jersey cows dramatically shifted in high THI conditions. More importantly, the hierarchical clustering of significant genes represented distinct gene expression changes between Holstein and Jersey PBMCs. These findings encouraged us to investigate the details of DEGs in Holstein and Jersey PBMCs between normal and hot and humid environmental conditions.

The overall significance and reliability of an animal study is sometimes limited by the number of animals. In particular, meta-transcriptome studies often draw conclusions from small numbers of samples. Although we also used a small sample of animals (*n* = 5 for each group) for transcriptome analysis, we observed highly clustered gene expression patterns in PCoA and heat map analysis. This indicates that the samples in each group had small variations thus, identified DEGs were significant.

In the top ten up- and down-regulated DEGs of dairy cows, we observed several significant DEGs that were associated with phagocytic function. Interestingly, different genes associated with phagocytic activity were down-regulated by heat stress in Holstein and Jersey cows. Under HT, the *DAB2* gene was significantly decreased in PBMCs in Holstein cow by heat stress. It has been reported that DAB2 is a specific transcription factor involved in phagocytosis in bovine classical monocytes [39]. In heat-stressed Jersey cows, the *GPR84* and *EDN1* gene expressions were significantly decreased. The GPR84 functions as a medium-chain, free fatty acid receptor and is expressed in immune cells, such as monocytes [40]. This gene up-regulation is known to enhance pro-inflammatory cytokine and chemokine expression in neutrophils, macrophages, and monocytes after stimulation, such as lipopolysaccharide (LPS) challenge [41,42]. Activation of the immune-metabolic receptor, GPR84, enhances phagocytosis and chemotaxis in macrophages [42,43]. The *EDN1* gene was the second gene in Jersey PBMCs that was significantly decreased by heat stress; it is one of the three isoforms of endothelin and is the most studied. Endothelin-1 regulates important phagocytic activities such as the trafficking of neutrophils and monocytes by controlling cytokine and chemoattractant [44]. Collectively, gene expression of *DAB2*, *GPR84*, and *EDN1* decreased in PBMCs by heat stress; these genes are usually important in phagocytic activities but have diverse functions in immune regulation. In vitro studies have also demonstrated that heat stress decreases the phagocytic activities of immune cells. Under heat stress, the phagocytic activity of peripheral monocytes from humans is reduced after LPS stimulation [45]. In addition, decreased phagocytosis in bovine immune cells under hot conditions was demonstrated by using the opsonization of fluorescein-labeled *Escherichia coli* [46]. Our findings, along with previous reports, strongly suggest that heat-stressed dairy cows may be highly susceptible to pathogenic infections due to impaired phagocytic activities concomitant with the down-regulation of key genes. However, direct functional comparisons between the phagocytic activities of phagocytes isolated from dairy cows under normal and heat-stress conditions need to be further investigated.

In the DEG analysis, we also observed that several chemokine genes were down-regulated by the HT season. A chemokine is a small, secreted protein released by cells and plays a pivotal role in the regulation of chemotaxis [47,48]. The chemokine family includes several ligands that bind to a smaller number of receptors, and they are key players in many immune responses such as inflammation, autoimmune, and infectious diseases [49]. In heat-stressed Holstein and Jersey cows, the expression of chemokine genes (*CXCL2* and *CXCL3*) was commonly decreased in both Holstein and Jersey PBMCs, in addition, *CXCL3* was the largest reduction in Jersey dairy cows. The chemokine, CXCL2, is produced by a variety of cell types such as macrophages, epithelial cells, hepatocytes, and monocytes in response to infection or injury. It assists in the recruitment of polymorphonuclear leukocytes (PMN) at the site of infection or injury and modulates immune responses [50]. The main function of CXCL3 is neutrophil trafficking, and for this reason, it is classified as an inflammatory chemokine [51]. There is an interesting study that newborn calves from heat-stressed cows have weaker neutrophil functions, such as phagocytosis and oxidative burst [52]. In dairy cow studies, the expression of *CXCL2* and *CXCL3* increased when treated with LPS or lipoteichoic acid (LTA), which are derived from major mastitis pathogens such as *E. coli* and *Staphylococcus aureus* in bovine, mammary, epithelial cells [53,54]. At high environmental temperatures, an intramammary infection challenge study using *Streptococcus uberis* demonstrated various immune responses [55]. This study showed altered neutrophil response-related gene expression. A more comprehensive study reported the antimicrobial activities of various chemokines against pathogens, including *E. coli* [56]. In the study, CXCL2 and CXCL3 were potent against *E. coli*, indicating their role in fighting pathogens in dairy cows. The weakened neutrophil functions concomitant with reduced *CXCL2* and *CXCL3* expression at high environmental temperatures may explain the increased infectious diseases such as mastitis in heat-stressed, dairy cows. 

In line with the observation that the DEGs are associated with chemokines, the KEGG pathway analysis revealed that many immune cell, migration-related pathways were significantly altered by heat stress. The top five KEGG pathways identified in both Holstein and Jersey PBMCs included the chemokine signaling pathway. Several chemokine signaling genes interacted in the gene network analysis of the dairy cows. The gene network was primarily populated with chemokine signaling pathways, and these genes were strongly associated with other genes. Abnormal immune cell migration as a result of altered chemokine expression may contribute to increased susceptibility to infectious diseases in dairy cows because immune cells are not able to migrate efficiently to infected sites, resulting in failed interaction with various immune cells. Studies on the role of chemokines in antimicrobial immunity in dairy cows are not available, but studies in other species suggest that reduced chemokine expression affects antimicrobial immunity [57]. Thus, abnormally reduced chemokine expression may be a reason for the attenuated immunity of heat-stressed dairy cows.

Another interesting immune pathway (and related genes) altered by THI condition was the cytokine response. In cytokine–cytokine receptor interaction, *IL1A* and *LOC529792* were commonly altered genes in both Holstein and Jersey PBMCs. In the top five KEGG pathways, cytokine–cytokine receptor interaction and viral protein interaction with cytokine and cytokine receptors were identified as significantly different pathways in Holstein and Jersey PBMCs. IL-1α exists in secreted or membrane-bound form in many cell types, such as hematopoietic and non-hematopoietic cells. Its expression can be increased by various stimuli, such as inflammation and stress and it ensures host protection and pathogen control by driving cell-cell cross interaction [51,58]. Mehla et al. reported that *IL1A* gene expression increased in short-duration heat stress but decreased in the recovery phase [59]. IL-1β is expressed primarily in monocytes and macrophages [47], and its secretion is induced primarily in response to microbial molecules such as LPS. Both IL-1α and IL-1β are classified as pro-inflammatory signaling cytokines [48]. In a mouse and human model, IL-1 cytokines restricted intracellular bacterial growth [60]. These two cytokines suppress *Mycobacterium tuberculosis* growth by regulating TNF receptor superfamily (TNFR) signaling and caspase-3 activation in macrophages. This antimicrobial activity leads to apoptosis, which restricts *M. tuberculosis* replication by efferocytosis. Thus, abnormally decreased *IL1A* and *IL1B* may inefficient induce pro-inflammatory responses to fight against pathogens.

We further identified more breed-specific, cytokine response-related genes and pathways. *IL1R2*, *IL1B*, and *IL10* genes were additionally identified in Jersey PBMCs. In addition, TNF signaling and the IL-17 signaling pathway were significantly altered in only Jersey PBMCs. These two pathways have a common intersection at AP-1 and IκBα. Upon stimulation by LPS or cytokines, IκBα is phosphorylated and degraded, and the resulting complex is translocated into the nucleus where it binds to κB sites and forms nuclear factor-kappa B (NF-κB) [61,62]. Another transcription factor, AP-1, which is a complex composed of proteins belonging to the FOS family, needs to dimerize to support its binding to AP-1 recognition sites [63]. Although we observed somewhat mixed results with the reduction in gene expression in both positive and negative regulators for these pathways, the expressions of all gene products including *IL1B*, *GRO1*, *CXCL2* in these two pathways were reduced. Thus, we can conclude that heat stress generally reduces the IL-17 and the TNF signaling pathways in Jersey PBMCs.

IL-17 and TNF are pro-inflammatory cytokines. IL-17, the signature cytokine secreted by Th17 cells, plays an important role in host defense against extracellular bacterial and fungal infections. Also, IL-17 induces inflammatory cytokine and chemokine production via pro-inflammatory signaling, such as NF-κB signaling [64]. Although the role of the IL-17 signal in the regulation of immune responses in ruminants is not fully understood, there are many mouse studies that indicate that the reduced IL17 signal pathway is able to attenuate antibacterial immunity. Bovine respiratory syncytial virus and *Mannheimia haemolytica* were used to infect calves to demonstrate that IL-17 was derived from CD4+ T cells and γδ T cells in PBMCs [65]. TNF signaling in ruminants was identified by transcriptome analysis using goat PBMCs infected with bovine viral diarrhea virus-2 (BVDV), an important viral pathogen of ruminants [66]. BVDV infection induced alteration in the TNF signaling pathway and related genes (*TNF-a*, *NFKBIA*, *CCL5*, *TNFAIP3*, *EDN1*) in the goat. A previous study reported that the expression of two genes (*IL17* and *TNF-a*) in Holstein and Holstein × Jersey crossbred dairy cows decreased when cows were immunosuppressed, such as after calving [67]. This suggests that these signaling pathways are sensitive to immune-suppressive conditions. Thus, in heat-stressed Jersey cows, impaired responsiveness of PBMCs to IL-17 and TNF may cause the attenuation of antimicrobial immunity at high environmental temperatures.

Lymphocytes, especially T cells, express calcium ion channels such as voltage-gated calcium (CaV) and P2X purinoreceptor (P2X) channels. The major functions of the calcium channel in T cells are cytokine production, cell survival, and T cell subset differentiation [68,69]. Calcium signals globally influence intracellular signal pathways of immune cells, resulting in an altered immune function [69]. We found that the calcium signaling pathway with DEGs was the most significantly different KEGG pathway in Holstein PBMCs. It has been reported that calcium levels are decreased in dairy cows in a high-temperature environment [70]. In Holstein cows, PBMCs may increase calcium influx gene expression to uptake calcium at HT; although we did not compare calcium levels in normal and heat-stress conditions, different calcium levels may be a cause for this. Interestingly, we did not observe the calcium signaling pathway in the list of significantly altered KEGG pathway categories in Jersey PBMCs. Although we do not understand the role played by the calcium signaling pathway in the regulation of immunological responses, increased calcium influx may somehow be related to the altered gene expression and function in Holstein PBMCs.

In this study, there were commonly altered genes in the chemokine and cytokine response to heat stress. We believe that common DEGs are likely to be the more important genes sensitive to hyperthermia in dairy cows. We found a decreased expression of *GRO1*, *CXCL2*, *CXCL3*, and *IL1A*; these are all important genes for innate immunity during pathogen infection. It is well documented that the incidence of clinical mastitis in Holstein dairy cows caused by pathogens increases during higher temperature seasons [71]. Heat-stressed dairy cows are vulnerable to pathogens such as *S. uberis*, which is a major cause of mastitis [55]. In a mouse study, heat stress negatively impacted vaccine effectiveness. Meng et al. [72] reported that chronic heat stress inhibits vaccine responses through regulating immune cell functions. The underlying mechanism is that the heat stress induces elevation of regulatory T cells and immune-suppressive cytokines such as IL-10 and TGF-β, resulting in a suppression of adaptive immunity. Although we do not fully understand the connection between elevated disease incidence and altered immune profiles, identified genes and pathways suggest that the potential mechanism of increased disease susceptibility by heat stress occurs together with altered immune characteristics. However, Holstein and Jersey cows showed differential patterns in immune shift, and disease occurrence in these two breeds of dairy cow may be explained by different immunological mechanisms.

In this study, Jersey PBMCs showed more significant changes in gene expression of various immune responses, including innate immunity. This finding suggests that Jersey cows require additional attention to maintain health status in HT as they may be potentially more susceptible to pathogenic infections by an unbalanced immune system. Direct comparisons of immunity and disease resistance between Holstein and Jersey dairy cows is limited. Transcriptomic studies were conducted to characterize the immune profiles of different breeds of dairy cow. A transcriptome study of leukocytes of three breeds of dairy cow (including Holstein and Jersey cows) reported remarkable, breed-specific gene expression patterns in the Holstein and Jersey [36]. This study revealed that Holstein and Jersey cows had distinct immune characteristics in respect of leukocytes. In our study, we did not present data on the direct comparison between Holstein and Jersey cows as we focused on natural, functional changes of PBMCs in the two breeds of dairy cow in the two environments. However, we also found that two breeds of dairy cow had different gene expression patterns in PBMCs (not shown). Unfortunately, we could not fully address why Jersey cows showed more dramatic changes in immunity compared to Holstein cows. We speculate that the difference in genetic background significantly influences the distinct immune changes between Holstein and Jersey dairy cows.

Immune responses can be altered by various factors, including genetics, nutrition, environmental conditions, and aging [73]. For example, nutrition supports the functions of immune cells. An activated immune system will increase energy demand and can initiate responses effectively and rapidly only with a sufficient supply of nutrients [74,75]. We fed the same TMR diet to both Holstein and Jersey cows and did not observe any limitation in dietary intake in either breed during this study, suggesting that some observed immune changes may not be driven by selective dietary intake or nutrient deficiency in a heat-stressed environment. However, we could not fully rule out the possibility that heat-stressed dairy cows had metabolic changes as a result of altered rumen fermentation even though feed intake was similar for the two breeds of dairy cow. Recently, our group reported that heat stress induced changes in the rumen microbiome in both Holstein and Jersey dairy cows [76]. Accumulating studies have reported that metabolites have a significant impact on immune responses in animals [77]. Thus, metabolite profiles in the rumen and blood need to be assessed in future studies in order to determine the possibility that altered metabolites significantly affect immunity of the dairy cow in heat stress. Additionally, age, lactation number, and lactation period can significantly affect the immune system of dairy cows. We started our experiment using dairy cows with a similar number of lactations and DIM. We monitored natural changes in the immune profile of PBMCs from May under MT conditions, and to August in HT conditions. Although our animal experiment was conducted during the mid-lactation period, there was a three-month interval. Thus, we cannot fully exclude the effects of aging or different lactation period on immune profile in this experimental design. Thus, our study represents natural changes of immunity of two breeds of dairy cow in the mid-lactation period from normal to heat-stress environmental conditions. To better understand the direct impact of heat stress on immune responses, in future studies, a chamber system or electric blanket should be applied to induce heat stress [37,78]. In this study, we found several altered genes related to critical immune functions in dairy cows by high THI environmental condition. In general, heat stress impaired immune functions to fight against pathogens. To elucidate the functional role of heat stress on immune response, immune challenges such as vaccination or experimentally induced pathogenic infection could be applied in the experimental design to determine the specific antimicrobial immunity of dairy cows as a result of seasonal change. 

## 5. Conclusions

The effects of THI condition (Moderate THI vs. High THI) on immunological response in Holstein and Jersey cows were examined by analyzing their PBMC transcriptome. The major finding of this study was the identification of heat-stress-induced, common and breed-specific DEGs through two breeds of dairy cow transcriptome analysis. The most prominent of specific genes from each transcriptome analysis are associated with immunological functions such as phagocytosis, chemokine, and cytokine response. Under the HT environment, we further identified five commonly altered genes (*CCR3*, *CXCL2*, *CXCL3*, *IL1A*, and *LOC529792*) in each dairy cow transcriptome, and they were related in chemokine and cytokine pathways among top five KEGG pathways. We also identified breed-specific pathways and related to DEGs. In the Holstein cows, the calcium signaling pathway (*CACNA1D*, *CACNA1E*, *CXCR4*, *MYLK*, *P2RX1*, *PPIF*, and *PTAFR*) and the cGMP-PKG signaling pathway (*CACNA1D*, *MYLK*, *PPIF*, *RGS2* (24 kDa), and *RGS2*) were represented, and the most of these gene expressions increased by heat-stress condition. The TNF signaling pathway (*CXCL2*, *CXCL3*, *EDN1*, *FOS*, *GRO1*, *ICAM1*, *IL1B*, *NFKBIA*, *PIK3R3*, *PTGS2*, and *TNFAIP3*) and the IL-17 signaling pathway (*CXCL2*, *CXCL3*, *FOS*, *FOSB*, *GRO1*, *IL1B*, *NFKBIA*, *PTGS2*, and *TNFAIP3*) were Jersey-specific pathways, and the most of these genes were down-regulated by high temperature and humid weather. Our transcriptome study provides key information concerning the common and breed-specific immunological changes in two breeds of dairy cow by high THI environment. These results bring a greater understanding of the mechanisms involved in the dampening of immune function in dairy cows and hence, potentially to enhance the productivity of these two globally important breeds.

## Figures and Tables

**Figure 1 animals-11-00019-f001:**
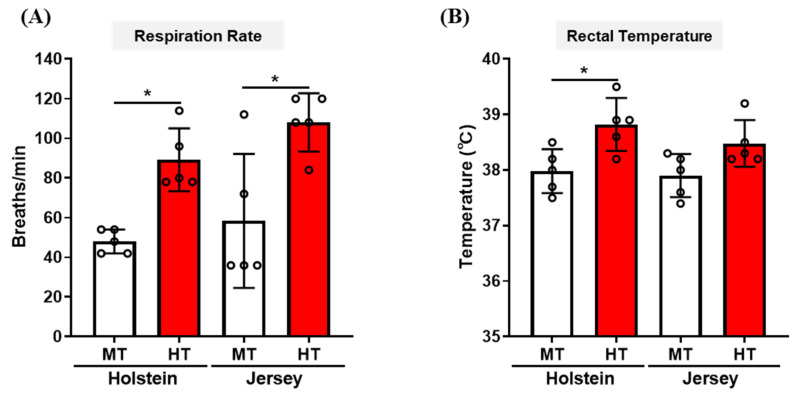
Physiological responses of Holstein and Jersey cows to differential seasonal environment. Measurements of respiration rate (**A**) and rectal temperature (**B**) for Holstein and Jersey cows in MT condition (THI = 69.6) and HT condition (THI = 87.5). Data are represented as mean ± SD; *n* = 5 animals/group. Values were statistically analyzed by repeated measures two-way ANOVA with Tukey’s multiple comparison test. * *p* < 0.05; MT, moderate THI season; HT, high THI season.

**Figure 2 animals-11-00019-f002:**
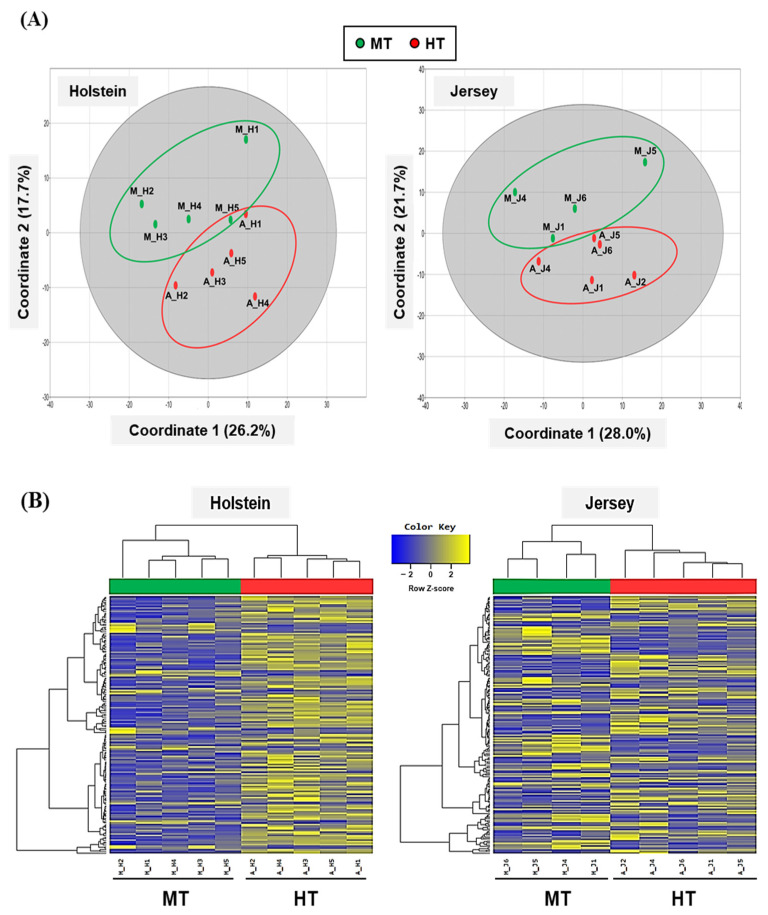
Differentially expressed genes in PBMCs of different breeds of dairy cow in different THI conditions. Principal coordinate analysis (PCoA) plots (**A**) and hierarchical clustering (**B**) of DEGs in PBMCs of Holstein and Jersey cows. MT, moderate THI season; HT, high THI season.

**Figure 3 animals-11-00019-f003:**
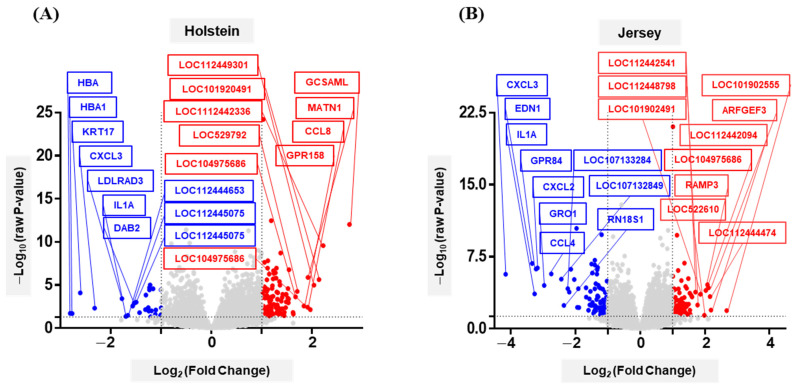
Volcano plots showing heat-stress-responding differentially expressed genes in Holstein and Jersey PBMCs. The DEGs in Holstein (**A**) and Jersey (**B**) PBMCs are presented in volcano plots (*p* < 0.05, fold-change < −2 or > 2). The top ten down-regulated (blue) or up-regulated (red) genes are highlighted.

**Figure 4 animals-11-00019-f004:**
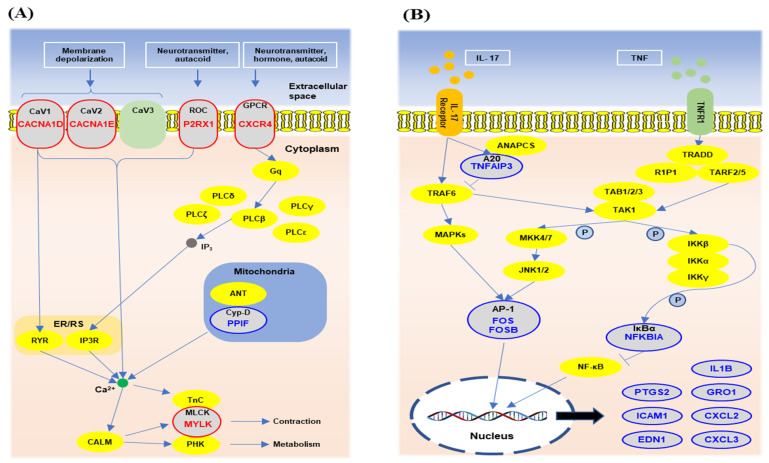
Top-ranked and interested pathways in Holstein and Jersey cows. Among the top five ranked KEGG pathways, the predominant pathway in Holstein and Jersey PBMCs is shown. The calcium signaling pathway of Holstein cows (**A**) and the IL-17 and TNF signaling pathways of Jersey cows (**B**) are presented with identified pathway associated genes. Blue and red colors indicate down-regulation and up-regulation, respectively.

**Figure 5 animals-11-00019-f005:**
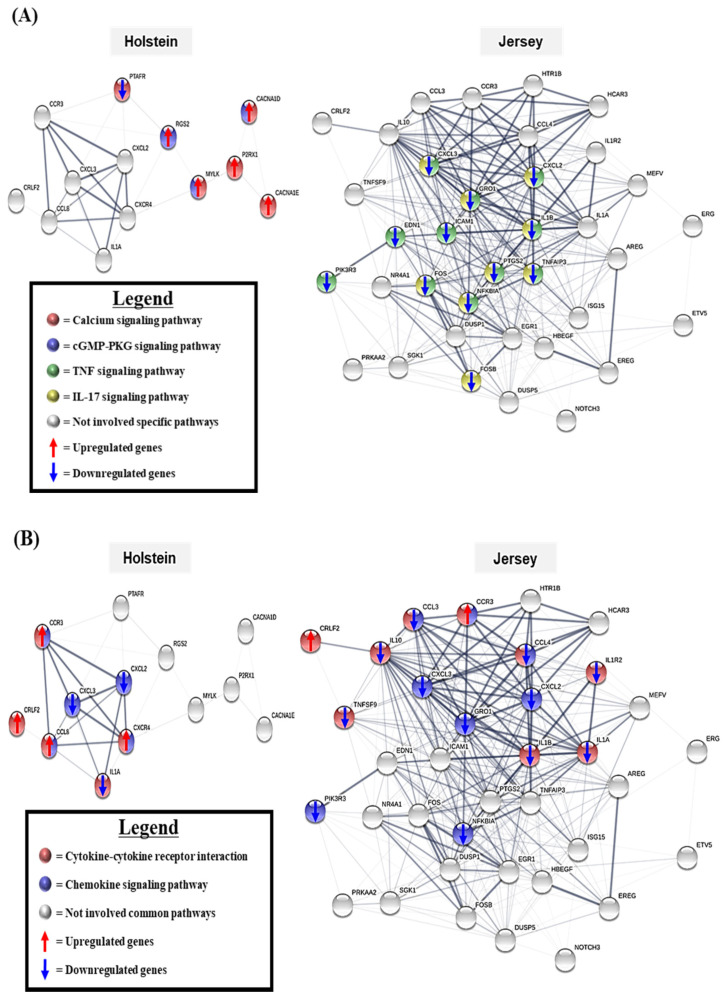
Gene network analysis of differentially expressed genes in predominant pathways. Gene network analysis was performed with the top five significant KEGG pathway-related DEGs of each dairy cow breed. The breed-specific pathway gene networks (**A**) and common pathway-related gene networks (**B**). The red-up and blue-down arrows indicate the up-regulation and down-regulation of specific genes by heat stress. The line thickness indicates the strength of edge confidence.

**Figure 6 animals-11-00019-f006:**
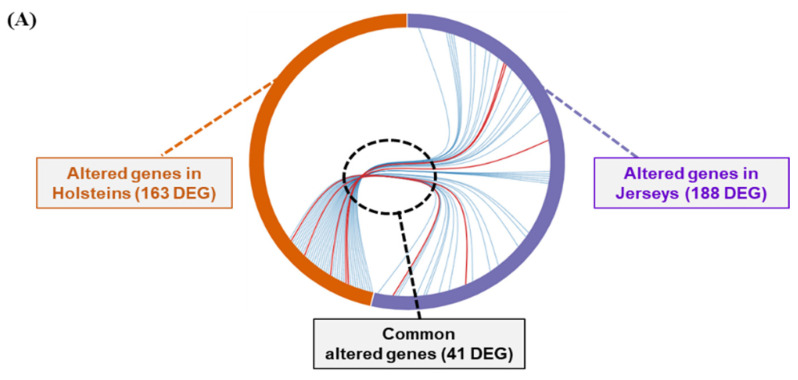
Summary of transcriptome analysis of PBMCs for two breeds of dairy cow in different THI conditions. Numbers of DEGs in Holstein and Jersey PBMCs in MT and HT condition (**A**). Heatmap represents a list of top five KEGG pathways and common and breed-specific DEGs-based genes associated with cut-off KEGG pathways (**B**).

## Data Availability

The data presented in this study are available on request from the corresponding author.

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
