# Peer review of "Common and Differential Dynamics of the Function of Peripheral Blood Mononuclear Cells between Holstein and Jersey Cows in Heat-Stress Environment"

_animals, 2020, doi:10.3390/ani11010019_

Round 1
Reviewer 1 Report
The study targets a current important issue in animal breeding and genetics, given that innate adaptation potential may play a fundamental role in response to changed climatic (and farming) condition.
The work is well organised and the paper is nicely written.
The overall significance and robustness of the study is somehow limited by the number of animals considered (although similar to other transcriptomic studies). Phenotypic data on response to heat stress (i.e. rectal temperature, decrease in production) did not yield significant differences, probably due to small sample size.
Most important, differential expression may by other confounding factors, stage of lactation in primis, that may have affected the results and should be discussed further in the manuscript.
As a curiosity, line 61, I don't understand the meaning of: immunity to vaccination.
Author Response
Response to Reviewer-1
The study targets a current important issue in animal breeding and genetics, given that innate adaptation potential may play a fundamental role in response to changed climatic (and farming) condition. The work is well organized and the paper is nicely written.
|
Point-1 |
The overall significance and robustness of the study is somehow limited by the number of animals considered (although similar to other transcriptomic studies). Phenotypic data on response to heat stress (i.e. rectal temperature, decrease in production) did not yield significant differences, probably due to small sample size. Most important, differential expression may by other confounding factors, stage of lactation in primis, that may have affected the results and should be discussed further in the manuscript.
Response: Thanks for valuable comments. We agree that sample size is critical factor to determine the phenotype particularly for in vivo study. Unfortunately, we utilized limited numbers of animals (5 animals for each Holstein and Jersey cows) in this study because 1) to start experiment with animals have similar lactation stage, 2) Feasibility of transcriptome analysis for PBMC samples. In fact, we had total 20 samples from Holstein and Jersey cows with normal and heat stress conditions. Although, animal numbers are a quite small however, we believe that phenotypes of Holstein and Jersey cows shown in our manuscript are still meaningful because 1) Although, we did show physiological responses of Holstein (n=5) and Jersey cows (n=5) in current manuscript, results from total eight animals (Five experimental animals used in our manuscript and three animals in same environmental condition) showed similar phenotype. As we performed RNA-seq analysis for five animals, we shown data for physiological responses from only five animals, 2) As we shown PCoA plots in Fig. 2A, samples from each groups were highly clustered, that indicating small variations within the groups. We added this point in revised manuscript (Line 408-412).
Also, we totally agree that identified DEGs in PBMCs can be induced by other confounding factors such as lactation stage. So, we added discussion for potential confounding factors in this study (Line 569-585). Collectively, we discussed limitation of our study such as potential compounding factors and numbers of animals in discussion part.
|
Point-2 |
As a curiosity, line 61, I don't understand the meaning of: immunity to vaccination.
Response: To make clearer, we have edited this sentence in revised manuscript as “For example, Heat-stressed dairy cows show abnormal innate and adaptive immune response, results in increased susceptibility of infectious diseases and dampened vaccine response.” (Line 64-66)

Reviewer 2 Report
Comments on Manuscript: Animals 995477
The aim of the present study was to assess if high temperature and humid weather could induce changes of specific gene expression and immune-pathways of peripheral blood mononuclear cells of Holstein and Jersey dairy cows.
The obtained results based on transcriptome analysis indicated a specific gene expression and immune-pathways from peripheral blood mononuclear cells of Holstein and Jersey dairy cows and showed that the two breeds have common and different immune responses to the heat stress.
General impression:
The topic in interesting for this field of research and contributes to further elucidate the immune responses of Holstein and Jersey cows to environmental change.
Furthermore, the manuscript is well written, the procedures and methods are well documented and the results are clearly presented.
Though, the introduction and discussion part must be improved. In the introduction part it would be appropriate to add data and information from the literature on susceptibility to pathologies and mortality relative to the two investigated breeds. In the discussion part it would be advisable to speculate more on the physiological interpretation and justification of the results since the Jersey cows showed greater impairment of the immune response than the Friesian cows despite being considered a more thermotolerant breed.
Comments and suggestion for improvement:
Line 94 : Please indicate the days in milK and the number of lactations of the dairy cows. Line 99: Explain the sentence better “there were no orts from any of the dairy cows”. Line 199-202: These sentences are considerations that must be included in the discussion and not in the description of the results and must be eliminated if already present because they are a repetition. Line 237-239: These sentences are considerations that must be included in the discussion and not in the description of the results and must be eliminated if already present because they are a repetition. Line 398: Please delete “figure 2” from the sentence.
Author Response
Response to Reviewer-2
The aim of the present study was to assess if high temperature and humid weather could induce changes of specific gene expression and immune-pathways of peripheral blood mononuclear cells of Holstein and Jersey dairy cows. The obtained results based on transcriptome analysis indicated a specific gene expression and immune-pathways from peripheral blood mononuclear cells of Holstein and Jersey dairy cows and showed that the two breeds have common and different immune responses to the heat stress.
General impression:
The topic in interesting for this field of research and contributes to further elucidate the immune responses of Holstein and Jersey cows to environmental change. Furthermore, the manuscript is well written, the procedures and methods are well documented and the results are clearly presented.
|
Point-1 |
Though, the introduction and discussion part must be improved. In the introduction part it would be appropriate to add data and information from the literature on susceptibility to pathologies and mortality relative to the two investigated breeds. In the discussion part it would be advisable to speculate more on the physiological interpretation and justification of the results since the Jersey cows showed greater impairment of the immune response than the Friesian cows despite being considered a more thermotolerant breed.
Response: Thanks for comments. To improve our manuscript, we have revised significant amounts of contents in both introduction and discussion part.
1) Introduction: Although, information on direct comparison of immunity or disease resistance/susceptibility between Holstein and Jersey cows are limited, we tried to add more relevant information in introduction part (Line 78-86).
2) Discussion: Unfortunately, we could not fully address the reason why Jersey cows showed more dramatic changes in immunity compared with Holstein cows, we have added more relevant discussions in revised manuscripts (Line 548-559). We believe that different genetic background significantly affects pattern of immune shift in two breeds of cows. It has been reported that Holstein and Jersey cows in normal condition had distinct immune characteristics [Huang et al., 2012].
Huang, W.; Nadeem, A.; Zhang, B.; Babar, M.; Soller, M.; Khatib, H. Characterization and comparison of the leukocyte transcriptomes of three cattle breeds. PLoS One. 2012, 7, e30244.
We also observed distinct gene expression pattern in PBMCs between Holstein and Jersey cows. As we want to focus on natural changes of immunity in each Holstein and Jersey by heat stress, we did not include this direct comparison in this manuscript.
|
Point-2 |
Line 94: Please indicate the days in milk and the number of lactations of the dairy cows.
Response: We added information in revised manuscript (Line 104-105; 109-110).
|
Point-3 |
Line 99: Explain the sentence better “there were no orts from any of the dairy cows”.
Response: In this experiment, the dairy cows were offered total of 20 kg/d (TMR) for each cow and there were no residuals in all experimental dairy cows. To make clear, we revised this sentence as “there was no residual TMR in any of the dairy cows.” (Line 119).
|
Point-4 |
Line 199-202: These sentences are considerations that must be included in the discussion and not in the description of the results and must be eliminated if already present because they are a repetition.
Line 237-239: These sentences are considerations that must be included in the discussion and not in the description of the results and must be eliminated if already present because they are a repetition.
Line 398: Please delete “figure 2” from the sentence.
Response: Thanks for suggestion. In revised manuscript, we have removed mentioned parts.

Reviewer 3 Report
Peer review report 1 on „Common and Differential Dynamics of the Function of Peripheral Blood Mononuclear Cells between Holstein and Jersey Cows in Heat stress Environment “
- Original Submission
- Recommendation
Major revision
- Comments to Author:
Manuscript number: 995477
Title: Common and Differential Dynamics of the Function of Peripheral Blood Mononuclear Cells between Holstein and Jersey Cows in Heat stress Environment
Eun Tae Kim, Sang Seok Joo, Dong Hyeon Kim, Bon-Hee Gu, Da Som Park, Rahman Md Atikur, Jun Kyu Son1, Beom Young Park, Sang Bum Kim, Tai-Young Hur, and Myunghoo Kim
Summer periods with high ambient temperatures are the consequence of the climate change jeopardizing animal welfare and health. Dairy cows may not well adapt to high ambient temperatures and are susceptible to health problems, e.g. viral infections, metabolic diseases due to a dysbalance of the immune system. Holstein and Jersey cows possess distinct breed characteristics with a higher milk yield of Holstein cows and higher milk fat and protein content of Jersey cows. Furthermore, Jersey cows have a greater capacity to adapt to high ambient temperature seen by lower rectal temperature. However, the adaptational mechanisms of the immune response to high ambient temperature were not studied in PBMCs of Holstein and Jersey cows. Transcript proofing of PBMCs was performed to investigate the environmental challenge comparing May and August blood samples from Holstein and Jersey cows. The cytokine and chemokine responses were the most prominent regulated immune pathways by high ambient temperatures. The manuscript provides new insides of the adaptational capacity of the immune system to environmental heat from two of the most important dairy breeds.
- Major comments:
- Originality/Novelty: The aim of this study was to identify common and breed-specific alterations of the immune response in Holstein and Jersey cows. Although the approach is new in comparing immune cells of two time point of two breeds and performing RNAseq, I see difficulties in the linguistic presented direct comparison of both breeds. The results refer only the comparison of one breed of transcript data from May compared with August, respectively. The presented conclusion draw out of this approach it is not possible. The direct comparison of both breeds from the time point August would be also no alternative due to the difference in the genetic background. The language must be wise chosen to differentiate between the RNAseq results from both breeds and requires revision. However, I do agree with the presentation of the data in tables and figures. Furthermore, all figures need to be enlarged. As an example, I highlight Fig. 5, where the legend and gene name are not clear readable. Please make sure, that size of the letters is between 7-9 in the final pdf version before uploading.
- Significance: L467-470 “Although Jerseys showed milder heat-stress related symptoms, interestingly, Jersey PBMC showed more significant changes in gene expression of various immune responses, including innate immunity. This finding suggests that Jersey cows required additional attention to maintain health status in HT as they may be potentially more susceptible to pathogenic infections.” Is there any evidence for this hypothesis that can be supported by the literature? How is that result linked to the fact that Jersey cows have a greater capacity to adapt to heat? Is the down regulation of the immune response in Jersey cows after heat stress not a sign for no activation? Based on the results presented, the hypothesis of a higher susceptibility to pathogenic infections is too speculative.
My recommendation is to carefully compare your data with the literature, e.g. Characterization and comparison of the leukocyte transcriptomes of three cattle breeds, Wen Huang , Asif Nadeem, Bao Zhang, Masroor Babar, Morris Soller, Hasan Khatib, PLoS One. 2012;7(1):e30244. doi: 10.1371/journal.pone.0030244. PMID: 22291923
L490: One of the main findings of heat-stressed Holstein cows is that the calcium signaling is activated. You speculate about the upregulation of related genes in Holstein PBMCs after heat stress. Could you be more precis and show evidence for upregulation of cytokine production, cell survival and T cell differentiation by specific CDs that are related to the differentiation, as mentioned in L489? Is there any evidence for higher cytokine production in your study based on serum or plasma analysis?
According to the supplemental data, CXCL3 in Jersey PBMC showed -17.8-fold change. Why is this fact not mentioned? What is the interpretation of this finding? In L432, the function of CXCL3 is described, but not the meaning in the heat stress context.
- Quality of Presentation: The presentation of the data in the results part needs revision – first: clarify the two different groups – Jersey and Holstein, and later breed differences. Second: if you mention 10 up- and 10 down-regulated genes, please make sure that 10 up- and down- regulated genes are mentioned in the result part (L249 – 8/10 down-regulated genes in Holstein cows, L251 – 7/10 up-regulated genes in Holstein cows, L257 – 7/10 up-regulated in Jersey cows). Third: what show the reader do with information that among the 10 up-regulated genes are 5 non-annotated? More information that is useful could be taken from those with known function.
L254 – In the results part, please mention facts and do not utilize filling sentences that are unnecessary.
Please add fold change in the results section of distinct genes that are from interest for further description in the later discussion part.
Discussion: The discussion part is hard to read and needs a revision. Beginning with L399, the top 10 different up- and down-regulated genes are presented, but this has not been consistently implemented for all 40 genes for both breeds. It is also not necessary to describe 40 genes, the most interesting findings for up- and down-regulation should be presented in here. Besides the description of the function of particular genes, I miss the interpretation of this finding in relationship to heat stress. As a result of top up- and down-regulated genes are the findings of the KEGG pathways. Do the results of the KEGG pathway needs a separate section? Could they not be described directly with up- and down-regulated genes?
Again, please make sure that the description of the differential dynamics in both breeds are clear as they are no direct comparison, as explained earlier. The outcome of the discussion should be precisely summarized in the conclusion and abstract part.
- Overall merit: The overall benefit of this transcriptome study is given by elucidating new adaptational mechanism of the immune system to heat stress in dairy cows, that was so far not investigated. This work provides new evidence for potentially treatment strategies to maintain animal health and reduce economical loss by ameliorating animal performance.
- Minor comments:
I recommend to the authors to have their paper language edited.
Page 1, Simple summary: L22 …and immune (response?). Word is missing.
Page 1, Abstract: L30 Why is heat stress bold? However,… - capital letter
Page 1, L33 immune responses – singular
Page 1, L35/36 … at moderate and high temperature-humidity index (THI) environmental conditions – rephrase this part, not clear.
Page 1, L38 – CXCL3 and IL1A in Holstein or Jesery cows?
Page 1, L43 – that changed by seasonal change – twice change
Page 2, L50 – again change..changes
Page 2, L52 - ..some of the most sensitive to heat stress – what? Missing word.
Page 2, L54 – economic losses – singular
Page 2, L57/58 – Literature is required.
Page 2, L71 – Please, precis the greater capacity to adapt to heat stress and why are Jersey cows able to adapt better to high ambient temperatures.
Page 3, L90 – description of the cows: please add days in milk, lactation period, pregnant or not, parity.
Page 3, L99 - … and there were no orts from any of the dairy cows. – what does that mean?
Page 3, L114 – At what time were blood samples taken? Morning? 14 o´clock as the THI was recorded?
Page 3, L120 – At what time point were respiration rate and rectal temperature measured?
Page 3, L124/125 – simply K2-EDTA vacutainer tubes - not a plastic whole blood tube, spray-coated
Page 4, L156 – what kind of paired-end sequencing was performed? 2x 100 bp sequencing cycle?
Page 5, L187 – Why did you change your statistical model? Instead of keeping proc mixed for repeated measurement as you did in
Differential Dynamics of the Ruminal Microbiome of Jersey Cows in a Heat Stress Environment. Kim DH, Kim MH, Kim SB, Son JK, Lee JH, Joo SS, Gu BH, Park T, Park BY, Kim ET. Animals (Basel). 2020 Jul 2;10(7):1127. doi: 10.3390/ani10071127. PMID: 32630754
Page 6, Figure 1 – A larger figure is required, only significant results should be presented, no ns in the figure.
Page 9, Figure 4 – Please provide a higher resolution of the figure.
Page 10, Figure 5 - Please provide a higher resolution of the figure.
Page 13, L447/448 – why LOC529792 was presented with gene name in brackets, but no other LOCs? That would be easier to read and to understand.
Page 15, L523 – why RGS2 with 25kDa in brackets? Please be consistent in your description.
Author Response
Response to Reviewer-3
Summer periods with high ambient temperatures are the consequence of the climate change jeopardizing animal welfare and health. Dairy cows may not well adapt to high ambient temperatures and are susceptible to health problems, e.g. viral infections, metabolic diseases due to a dysbalance of the immune system. Holstein and Jersey cows possess distinct breed characteristics with a higher milk yield of Holstein cows and higher milk fat and protein content of Jersey cows. Furthermore, Jersey cows have a greater capacity to adapt to high ambient temperature seen by lower rectal temperature. However, the adaptational mechanisms of the immune response to high ambient temperature were not studied in PBMCs of Holstein and Jersey cows. Transcript proofing of PBMCs was performed to investigate the environmental challenge comparing May and August blood samples from Holstein and Jersey cows. The cytokine and chemokine responses were the most prominent regulated immune pathways by high ambient temperatures. The manuscript provides new insides of the adaptational capacity of the immune system to environmental heat from two of the most important dairy breeds.
- Major comments:
|
Point-1 |
Originality/Novelty: The aim of this study was to identify common and breed-specific alterations of the immune response in Holstein and Jersey cows. Although the approach is new in comparing immune cells of two time point of two breeds and performing RNAseq, I see difficulties in the linguistic presented direct comparison of both breeds. The results refer only the comparison of one breed of transcript data from May compared with August, respectively. The presented conclusion draw out of this approach it is not possible. The direct comparison of both breeds from the time point August would be also no alternative due to the difference in the genetic background. The language must be wise chosen to differentiate between the RNAseq results from both breeds and requires revision. However, I do agree with the presentation of the data in tables and figures. Furthermore, all figures need to be enlarged. As an example, I highlight Fig. 5, where the legend and gene name are not clear readable. Please make sure, that size of the letters is between 7-9 in the final pdf version before uploading.
Response: Thanks for valuable comments. We totally understand your opinion that there is a limitation to convey all information from RNA-seq results through current version of manuscript. We have total four different groups; Holstein (May), Holstein (Aug), Jersey (May), and Jersey (Aug). So, it is possible to make many different paired-comparisons such as Holstein (Aug) vs. Jersey (Aug) as you mentioned. After deep consideration, we have decided to focus on comparison in gene expression pattern of each Holstein and Jersey cows in two different time points (normal vs. heat stress). We thought that this is the best way to show natural changes of immunity of each two breeds of dairy cow by seasonal change and that is really occur in field of dairy industry. However, we revised manuscript to clearly convey the information from RNA-seq results.
1) We have added sentence to emphasize the way of comparison and analysis in part of abstract (Line 36-39) and introduction (Line 89-92).
2) We went through whole manuscript and edited words to make clearer. For example, “DEG analysis was performed to compare the differential gene expressions of each Holstein and Jersey cows between HT and MT condition.” (Line 234-236)
3) We changed size and resolution of all figures with minor corrections. We believe that revised manuscript is now in better shape to present our findings.
|
Point-2 |
Significance: L467-470 “Although Jerseys showed milder heat-stress related symptoms, interestingly, Jersey PBMC showed more significant changes in gene expression of various immune responses, including innate immunity. This finding suggests that Jersey cows required additional attention to maintain health status in HT as they may be potentially more susceptible to pathogenic infections.” Is there any evidence for this hypothesis that can be supported by the literature? How is that result linked to the fact that Jersey cows have a greater capacity to adapt to heat? Is the down regulation of the immune response in Jersey cows after heat stress not a sign for no activation? Based on the results presented, the hypothesis of a higher susceptibility to pathogenic infections is too speculative. My recommendation is to carefully compare your data with the literature, e.g. Characterization and comparison of the leukocyte transcriptomes of three cattle breeds, Wen Huang , Asif Nadeem, Bao Zhang, Masroor Babar, Morris Soller, Hasan Khatib, PLoS One. 2012;7(1):e30244. doi: 10.1371/journal.pone.0030244. PMID: 22291923
Response: Thanks for valuable comment and suggestion. Actually, it is well documented that incidence of clinical mastitis caused by pathogens such as staphylococcus aureus and Escherichia coli increased during high temperature season. Although, studies on heat stress-induced alteration of immunity in dairy cows are limited, it was reported that in mouse study, chronic heat stress inhibits immune responses to virus infection through suppressed immune function. Also, identified DEGs (eg., cytokine and chemokine response) are key immune factors to fight against pathogens. Thus, abnormally decreased cytokine/chemokine-related pathways by heat stress condition are able to make animals are inefficiently respond to pathogens. We added these points in revised manuscript with more references (Line 535-547).
Thompson, I.; Tao, S.; Monteiro, A.; Jeong, K.; Dahl, G.J. Effect of cooling during the dry period on immune response after Streptococcus uberis intramammary infection challenge of dairy cows. J. Dairy Sci. 2014, 97, 7426-7436.
Riekerink, R.O.; Barkema, H.; Stryhn, H. The effect of season on somatic cell count and the incidence of clinical mastitis. J. Dairy Sci. 2007, 90, 1704-1715.
Meng, D.; Hu, Y.; Xiao, C.; Wei, T.; Zou, Q.; Wang, M. Chronic heat stress inhibits immune responses to H5N1 vaccination through regulating CD4+ CD25+ Foxp3+ tregs. Biomed Res. Int. 2013, 2013.
|
Point-3 |
L490: One of the main findings of heat-stressed Holstein cows is that the calcium signaling is activated. You speculate about the upregulation of related genes in Holstein PBMCs after heat stress. Could you be more precis and show evidence for upregulation of cytokine production, cell survival and T cell differentiation by specific CDs that are related to the differentiation, as mentioned in L489? Is there any evidence for higher cytokine production in your study based on serum or plasma analysis?
Response: Thanks for comments. As we discussed, Ca signal is important for immune regulation. However, increased Ca influx simply does not simply mean upregulation of pro-inflammatory cytokines. Also, we could not observe increased cytokine expression in Holstein cows with heat stress. As Ca signal globally influence on various signal pathways in immune cells, it is really hard to specify functional changes of immune cells. Please, understand that limited information is available. We carefully revised about this issue in discussion part. (Line 519-531)
|
Point-4 |
According to the supplemental data, CXCL3 in Jersey PBMC showed -17.8-fold change. Why is this fact not mentioned? What is the interpretation of this finding? In L432, the function of CXCL3 is described, but not the meaning in the heat stress context.
Response: Thanks for comments: We emphasized high fold changes of CXCL3 in result and discussion section of revised manuscript (Line 262-271). Also, we added more discussions about potential role of CXCL3 in regulation for immune regulation together with CXCL2 (Line 443-461).
|
Point-5 |
Quality of Presentation: The presentation of the data in the results part needs revision – first: clarify the two different groups – Jersey and Holstein, and later breed differences. Second: if you mention 10 up- and 10 down-regulated genes, please make sure that 10 up- and down- regulated genes are mentioned in the result part (L249 – 8/10 down-regulated genes in Holstein cows, L251 – 7/10 up-regulated genes in Holstein cows, L257 – 7/10 up-regulated in Jersey cows). Third: what show the reader do with information that among the 10 up-regulated genes are 5 non-annotated? More information that is useful could be taken from those with known function.
L254 – In the results part, please mention facts and do not utilize filling sentences that are unnecessary.
Please add fold change in the results section of distinct genes that are from interest for further description in the later discussion part.
Response: Thank you for kind suggestion. We agree that our results part need to be revised to clear present results for readers. We revised results parts as followed your suggestion.
1) We described results more systemically. For example, we have explained the two breeds of dairy cows first, then the differences between the Holstein and Jersey cows.
2) We added all 10 up- and 10- down regulated genes in the revised figure 3 and manuscript (Line 255-268).
3) We changed or removed interpreted language in discussion section.
4) We further discussed functional role of top up- or down- reregulated DEGs in revised manuscript.
5) We add fold change value of genes in result section up to second digit in the revised manuscript.
|
Point-6 |
Discussion: The discussion part is hard to read and needs a revision. Beginning with L399, the top 10 different up- and down-regulated genes are presented, but this has not been consistently implemented for all 40 genes for both breeds. It is also not necessary to describe 40 genes, the most interesting findings for up- and down-regulation should be presented in here. Besides the description of the function of particular genes, I miss the interpretation of this finding in relationship to heat stress. As a result of top up- and down-regulated genes are the findings of the KEGG pathways. Do the results of the KEGG pathway needs a separate section? Could they not be described directly with up- and down-regulated genes?
Again, please make sure that the description of the differential dynamics in both breeds are clear as they are no direct comparison, as explained earlier. The outcome of the discussion should be precisely summarized in the conclusion and abstract part.
Response: Thank you for comments and suggestion for discussion part. We tried to improve discussion section as followed your suggestions.
1) In our study, we found several genes and pathways, that altered by heat stress in each Holstein and Jersey cows. Because KEGG pathways had too broad category, we decided to discuss our findings with focusing on specific immunological factors; there functions are 1) phagocytosis, 2) chemokine, 3) cytokine response. We believe that this way is better to convey the immunological meanings from our data for readers. However, we did put more efforts on discussion part to have more clear messages from our experiment. Please, find revised manuscript.
2) Unfortunately, there are almost no reports on direct link between identified genes from our study and heat stress. Thus, we tried to explain their connections through discussing functional role of genes in immune responses and potential implications. Please find revised manuscript.
3) We have revised manuscript to emphasize our major finding that differential dynamics of immune response in two breeds of dairy cows by heat stress. Please, find significant changes in abstract and conclusion part of revised manuscript.
- Overall merit: The overall benefit of this transcriptome study is given by elucidating new adaptational mechanism of the immune system to heat stress in dairy cows, that was so far not investigated. This work provides new evidence for potentially treatment strategies to maintain animal health and reduce economical loss by ameliorating animal performance.
- Minor comments:
I recommend to the authors to have their paper language edited.
Page 1, Simple summary: L22 …and immune (response?). Word is missing.
Page 1, Abstract: L30 Why is heat stress bold? However,… - capital letter
Page 1, L33 immune responses – singular
Response: We apologized for minor errors. We revised them in the revised manuscript.
Page 1, L35/36 … at moderate and high temperature-humidity index (THI) environmental conditions – rephrase this part, not clear.
Response: We rephrase this part “… between moderate temperature-humidity index (THI) and high THI environmental condition.” (Line 38-39)
Page 1, L38 – CXCL3 and IL1A in Holstein or Jersey cows?
Response: Sorry for confusing. The CXCL3 and IL1A were decreased in both Holstein and Jersey cows. We revised sentence containing this part as “Among DEGs, CXCL3 and IL1A were top down-regulated genes in both breeds of dairy cow…” (Line 41-42)
Page 1, L43 – that changed by seasonal change – twice change
Page 2, L50 – again change..changes
Response: We revised this part as “that altered by THI environmental condition.” (Line 45-46) And We paraphrased “change” as other words in the revised manuscript (Line 53).
Page 2, L52 - ..some of the most sensitive to heat stress – what? Missing word.
Page 2, L54 – economic losses – singular
Response: Sorry for error. We edited these points.
Page 2, L57/58 – Literature is required.
Response: We added reference in the revised manuscript. (Line 60)
Gernand, E.; König, S.; Kipp, C. Influence of on-farm measurements for heat stress indicators on dairy cow productivity, female fertility, and health. J. Dairy Sci. 2019, 102, 6660-6671.
Page 2, L71 – Please, precis the greater capacity to adapt to heat stress and why are Jersey cows able to adapt better to high ambient temperatures.
Response: We added more information and modified related-contents in revised manuscript. We believe that massages can be presented more clearly to readers (Line 72-77).
Page 3, L90 – description of the cows: please add days in milk, lactation period, pregnant or not, parity.
Response: Thanks for comment. We added information in the text. (Line 104-105; 109-110)
Page 3, L99 - … and there were no orts from any of the dairy cows. – what does that mean?
Response: Sorry for confusing. In this experiment, the dairy cows were offered total of 20 kg/d (TMR) for each cow and there were no residuals in all experimental dairy cows. We edited this sentence in revised manuscript as “there was no residual TMR in any of the dairy cows.” (Line 119)
Page 3, L114 – At what time were blood samples taken? Morning? 14 o´clock as the THI was recorded?
Page 3, L120 – At what time point were respiration rate and rectal temperature measured?
Response: We have collected blood samples after recording THI (14:00) and measured respiration rate and rectal temperature before blood collection. We described details for sample collection in the revised manuscript. (Line 134; 138-139)
Page 3, L124/125 – simply K2-EDTA vacutainer tubes - not a plastic whole blood tube, spray-coated
Response: As followed your recommendation, we changed blood collection tube name more simply. (Line 141-142)
Page 4, L156 – what kind of paired-end sequencing was performed? 2x 100 bp sequencing cycle?
Response: We added information “paired-end (2 x 100 bp) sequencing” in the revised manuscript. (Line 173)
Page 5, L187 – Why did you change your statistical model? Instead of keeping proc mixed for repeated measurement as you did in Differential Dynamics of the Ruminal Microbiome of Jersey Cows in a Heat Stress Environment. Kim DH, Kim MH, Kim SB, Son JK, Lee JH, Joo SS, Gu BH, Park T, Park BY, Kim ET. Animals (Basel). 2020 Jul 2;10(7):1127. doi: 10.3390/ani10071127. PMID: 32630754
Response: We tried both repeated measure Two-way ANOVA and proc mixed for repeated measurement. We got same results from both statistical Data Analysis. We added additional statistical analysis for measurements used in our study (Line 202-204).
Page 6, Figure 1 – A larger figure is required, only significant results should be presented, no ns in the figure.
Page 9, Figure 4 – Please provide a higher resolution of the figure.
Page 10, Figure 5 - Please provide a higher resolution of the figure.
Response: Thanks for comments. We changed size and resolution for all figures in the revised manuscript.
Page 13, L447/448 – why LOC529792 was presented with gene name in brackets, but no other LOCs? That would be easier to read and to understand.
Response: Actually, The LOC529792 gene was presented as CRLF2 (Cytokine receptor-like factor 2) in STRING software (Figure 5). To make understanding easier, we revised CRLF2 as LOC529792 in the revised manuscript. However, in order to clarify in the figure 5, we have only given the gene name once in the results part (3.5. The gene network analysis of the unique differentially expressed genes).
Page 15, L523 – why RGS2 with 25kDa in brackets? Please be consistent in your description.
Response: In our study, two RGS2 genes were characterized in Holstein cows. The gene symbol of these genes was identical in Supplementary file 1. However, according to Entrez (http://www.ncbi.nlm.nih.gov/Entrez/), the full name of genes was presented as regulator of G protein signaling 2 (RGS2, 100848920 (Entrez ID)) and regulator of G protein signaling 2, 24kDa (RGS2, 513055 (Entrez ID)). To distinguish between these two genes, we presented RGS2 and RGS2 (24kDa) in the figure 6 and conclusion part.
